# Role of HMGB1 in an Animal Model of Vascular Cognitive Impairment Induced by Chronic Cerebral Hypoperfusion

**DOI:** 10.3390/ijms21062176

**Published:** 2020-03-21

**Authors:** Amelia Nur Vidyanti, Jia-Yu Hsieh, Kun-Ju Lin, Yao-Ching Fang, Ismail Setyopranoto, Chaur-Jong Hu

**Affiliations:** 1International PhD Program in Medicine, College of Medicine, Taipei Medical University, Taipei 11031, Taiwan; amelia.nur.v@ugm.ac.id; 2Department of Neurology, Faculty of Medicine, Public Health and Nursing, Universitas Gadjah Mada, Yogyakarta 55281, Indonesia; ismail.setyopranoto@ugm.ac.id; 3Department of Neurology, College of Medicine, Taipei Medical University, Taipei 11031, Taiwan; m120097049@tmu.edu.tw; 4Department of Nuclear Medicine and Molecular Imaging Center, Linkou Chang Gung Memorial Hospital, Taoyuan City, 33302, Taiwan; kunjulin@gmail.com; 5Healthy Aging Research Center and Department of Medical Imaging and Radiological Sciences, College of Medicine, Chang Gung University, Taoyuan City 33302, Taiwan; 6Taipei Neuroscience Institute, Taipei 23561, Taiwan; eugene_8888@yahoo.com.tw; 7Department of Neurology and Dementia Center, Shuang Ho Hospital, Taipei Medical University, New Taipei City 23561, Taiwan; 8The PhD Program for Neural Regenerative Medicine, College of Medical Science and Technology, Taipei Medical University, Taipei 11031, Taiwan

**Keywords:** vascular cognitive impairment, chronic cerebral hypoperfusion, modified bilateral common carotid artery occlusion, HMGB1

## Abstract

The pathophysiology of vascular cognitive impairment (VCI) is associated with chronic cerebral hypoperfusion (CCH). Increased high-mobility group box protein 1 (HMGB1), a nonhistone protein involved in injury and inflammation, has been established in the acute phase of CCH. However, the role of HMGB1 in the chronic phase of CCH remains unclear. We developed a novel animal model of CCH with a modified bilateral common carotid artery occlusion (BCCAO) in C57BL/6 mice. Cerebral blood flow (CBF) reduction, the expression of HMGB1 and its proinflammatory cytokines (tumor necrosis factor-alpha [TNF-α], interleukin [IL]-1β, and IL-6), and brain pathology were assessed. Furthermore, we evaluated the effect of HMGB1 suppression through bilateral intrahippocampus injection with the CRISPR/Cas9 knockout plasmid. Three months after CCH induction, CBF decreased to 30–50% with significant cognitive decline in BCCAO mice. The 7T-aMRI showed hippocampal atrophy, but amyloid positron imaging tomography showed nonsignificant amyloid-beta accumulation. Increased levels of HMGB1, TNF-α, IL-1β, and IL-6 were observed 3 months after BCCAO. HMGB1 suppression with CRISPR/Cas9 knockout plasmid restored TNF-α, IL-1β, and IL-6 and attenuated hippocampal atrophy and cognitive decline. We believe that HMGB1 plays a pivotal role in CCH-induced VCI pathophysiology and can be a potential therapeutic target of VCI.

## 1. Introduction

Vascular cognitive impairment (VCI) after Alzheimer’s disease (AD) is the second leading cause of dementia worldwide. VCI includes a clinical spectrum of cognitive decline, ranging from vascular mild cognitive impairment to vascular dementia. One major cause of cognitive decline in VCI is chronic cerebral hypoperfusion (CCH) [1]. CCH may develop in many conditions involving the impairment of the cerebral vascular system, such as hypotension, extreme hypertension, diabetes mellitus, atherosclerosis, smoking, and carotid or cerebral artery stenosis [2]. Animal models of CCH showed that a persistent decrease in cerebral blood flow (CBF) was followed by the production of reactive oxygen species and proinflammatory cytokines. Eventually, it may damage neuronal cells and lead to neurodegeneration [3,4,5,6]. Nevertheless, prior studies had some limitations regarding relatively high mortality rates and the visual pathway damage of rat models of CCH [7,8,9,10]; furthermore, microcoils or ameroid constrictors used for mice models were expensive [5,10,11,12].

Cerebral hypoperfusion causes neuronal damage by triggering at least two sequences of excitotoxicity [13,14]. The first sequence is related to the abrupt depletion of adenosine triphosphate (ATP) due to a sudden decrease in CBF. This ischemic condition causes neuronal necrosis in the acute stage. The second sequence involves gradual CBF restoration but moderate CBF reduction; the ATP level is still low, and this condition may result in neuronal apoptosis. However, neuronal death after CCH is not solely typical necrosis or apoptosis. Instead, it involves complex molecular and biochemical mechanisms along with the apoptosis–necrosis continuum [15,16].

Necrotic neurons release the high-mobility group box protein 1 (HMGB1). HMGB1 is a nonhistone protein that serves as a crucial cytokine produced in response to infection, injury, and inflammation. HMGB1 is normally located in the nucleus [17]. HMGB1 can be either actively secreted by inflammatory cells or passively released by necrotic or dead cells. Extracellularly located HMGB1 subsequently induces the inflammation cascade by binding to the toll-like receptor (TLR)4 or receptor for advanced glycation end products (RAGE) [18].

The role of HMGB1 has been widely established in acute stroke and traumatic brain injury because it is responsible for neuronal damage and symptom worsening. Moreover, several studies have been conducted on HMGB1-induced chronic inflammation and neurodegeneration in AD, amyotrophic lateral sclerosis, Parkinson disease, and multiple sclerosis. HMGB1 could impair neurite outgrowth in a mouse model of AD and lead to cognitive decline [19]. In a human study, serum HMGB1 increased significantly in AD patients and correlated well with the amyloid-β level [20]. Moreover, HMGB1 is believed to disrupt the blood–brain-barrier integrity [21].

The detailed mechanisms of HMGB1 in mediating cognitive decline at the chronic phase after CCH have not yet been established. An experimental model of CCH in a previous study demonstrated that the administration of anti-HMGB1 neutralizing antibody (Ab) at the acute stage of ischemia could reserve hippocampal neuronal death and improve cognitive impairment [22]. Although the positive effect of Ab remains for 12 weeks, the persistence of high HMGB1 at the chronic phase of CCH has not been investigated. In addition, it is unclear whether HMGB1 suppression in the chronic phase would be beneficial. This is the first study to use modified bilateral common carotid artery occlusion (BCCAO) to induce CCH in mice and suppress the HMGB1 level with the CRISPR/Cas9 knockout (KO) plasmid during the chronic stage. Hence, we aimed to determine the role of HMGB1 in the chronic stage of CCH and to explore its potential application in VCI.

## 2. Results

### 2.1. Survival Rate and CBF Values

All procedures of modified-BCCAO were accomplished within 15 min, except for a waiting time of 30 min to transiently ligate the left CCA. After 3 months of BCCAO surgery, all mice had survived.

In the control group (sham surgery), the mean CBF at 1 month and 3 months after sham surgery did not change much from the baseline. By contrast, CBF ratios decreased significantly in the BCCAO group. Compared with sham mice, the CBF ratios of BCCAO mice in the right and left cerebral cortex at 1 month after surgery were 57.5% ± 13.6% and 73.4% ± 15.5% (mean ± standard deviation), respectively (Figure 1A). At 3 months after surgery, CBF ratios remained low in BCCAO mice. They were 58.4% ± 3.3% and 69.9% ± 4.4% in the right and left cerebral cortex, respectively (Figure 1A). CBF changes in the BCCAO group were significantly different compared with those in the sham group (sham, *n* = 6; BCCAO, *n* = 6).

### 2.2. CCH Induced Hippocampal Atrophy and Memory Decline but Did Not Cause Behavioral Alterations in Motor Coordination

To investigate the effects of CCH on motor coordination, we performed the rotarod test and beam-balance walking test at 3 months after surgery (Figure 1B). We found that BCCAO did not alter motor coordination, indicating no injury in the cortical or subcortical lesion.

To study whether CCH impairs memory function, the novel object recognition (NOR) test was conducted. This test can be used to assess nonspatial working memory, which describes some parts of the hippocampus function and its relationship with the perirhinal cortex [23] as well as frontal subcortical circuits [24]. BCCAO mice showed poorer performance in recognizing familiar objects compared with sham mice at 3 months after surgery (Figure 1C; F (1, 10) = 24.05, *p* < 0.001). Compared with sham mice, BCCAO mice spent less time exploring the novel object when tested at 3 h after the training phase. Hence, the results of RI < 50% indicates a short-term memory deficit in BCCAO mice due to the impairment of memory retention for novel objects or more preference for familiar objects [25,26].

Hippocampal atrophy is associated with cognitive decline in dementia [27,28,29]. To delineate this, hippocampal volumes were measured in sham and BCCAO mice at 3 months after surgery by using T1- and T2-weighted images. Hippocampal atrophy was found during the time course after surgery in BCCAO mice but not in sham mice. BCCAO mice showed significant reduction of the total hippocampal volume than sham mice at 3 months after surgery (Figure 1D; *p* < 0.05).

CCH is expected to cause amyloid-beta (Aβ) accumulation, particularly in AD [30,31,32,33]. Furthermore, we used amyloid-PET scanning to investigate whether CCH induced by BCCAO could develop Aβ accumulation at 3 months after surgery. However, no Aβ accumulation was observed in BCCAO mice (Figure 2).

### 2.3. CCH Increased the Expression of HMGB1 and Its Proinflammatory Cytokines at 3 Months

CCH induces neuroinflammation [29,34]. However, numerous cascades of inflammation are involved in CCH. We observed that CCH induced by BCCAO altered the HMGB1 level in the cortex and hippocampus at 3 months (Figure 3A,B).

HMGB1 expression at the chronic phase of CCH was further confirmed through immunostaining (Figure 3C,D). We consistently found that HMGB1 was expressed more in BCCAO mice than in sham mice. Furthermore, proinflammatory cytokines, such as TNF-α and IL-1β, were upregulated (Figure 3C,D). In addition, IL-6 was likely to increase in BCCAO mice compared with sham mice but non-significantly (data not shown). All of them serve as the downstream signaling pathway of HMGB1.

### 2.4. Administration of CRISPR/Cas9-KO of HMGB1 in BCCAO Mice Reduces the Expression of HMGB1 and Its Proinflammatory Cytokines, Attenuates Hippocampal Atrophy, and Improves Cognitive Decline

To further delineate the role of HMGB1 at the chronic phase of CCH, we injected the HMGB1 CRISPR/Cas9 KO plasmid for suppressing HMGB1. The injection was administered 1 month after CCH because during this time, CCH begins the chronic phase, which leads to oligemia [6]. Two months after the injection, BCCAO HMGB1-KO mice showed a decreased level of HMGB1 protein, which was confirmed through Western blot and immunostaining (Figure 4A–D). Moreover, HMGB1 suppression could decrease proinflammatory cytokines (TNF-α, IL-1β, and IL-6), attenuate hippocampal atrophy, and improve cognitive decline (Figure 5).

## 3. Discussion

This is the first study to explore the role of HMGB1 at the chronic phase of CCH in a novel mouse model of VCI induced by modified-BCCAO. Moreover, this is the first study to use the HMG-1 CRISPR/Cas9 KO plasmid to suppress HMGB1 in CCH. We observed that HMGB1 increased 3 months after CCH along with a decreased CBF ratio from baseline, hippocampal neuronal loss or atrophy, and declined memory function. The administration of the HMG-1 CRISPR/Cas9 KO plasmid at 1 month after surgery reversed those consequences of CCH.

The present study demonstrated that CCH induced by modified-BCCAO surgery in mice caused moderate hypoperfusion at 3 months after surgery. The CBF ratio was persistently decreased to 50–70% from baseline even after 3 months. A previous study performed quantitative measurement through laser speckle imaging, which revealed that in a bilateral carotid artery stenosis model, the CBF decreased to 62.9% at 2 h after surgery and was gradually restored to 81.7% at 1 month, 83.2% at 2 months, and 85.0% at 3 months after surgery [5]. Another model of VCI, unilateral common carotid artery occlusion, has shown that CBF measured using laser Doppler flowmetry decreased to approximately 20–37% at 28 days after surgery [4,27]. To the best of our knowledge, this is the first study to show that the CBF ratio remains low at 3 months after surgery [5,6,11,35]. The results of the present study might be because the BCCAO procedure consists of two steps, which reduces death risk and causes persistently moderate reduction of CBF until 3 months after surgery.

In the present study, the low RI (<0.5) in the NOR test of the CCH model indicates significant impairment of nonspatial memory function. Moreover, other studies have shown similar findings [4,5,36,37]. The NOR test could demonstrate the essential function of the hippocampus (especially the dorsal part) and perirhinal cortex [38,39]. These two structures are responsible for nonspatial and short-term memory function [23,24]. This result was further supported by hippocampal atrophy on MRI. Hippocampus is the most vulnerable region after global cerebral ischemia or CCH, especially CA1 neurons. Therefore, hippocampal atrophy could be found as the consequence of neuronal loss after CCH [6,40,41,42]. We did not observe any motoric disturbance at 3 months after surgery, indicating that CCH does not alter the motor cortex or other structures associated with motoric function. This is in agreement with the findings of prior studies [4,32,43].

In the present study, we did not observe any Aβ accumulation. Thus, the CCH process might not be adequate to induce Aβ accumulation in C57BL/6 mice. Factors other than CCH are required to initiate Aβ deposition. Studies have reported that Aβ accumulation occurs in AD transgenic mice (aged 5 months) 1 month after CCH induction [32,36]. In another study, a 3-month-old AD transgenic mouse demonstrated Aβ overproduction, decreased α-secretase activity and expression, and increased β-secretase activity and expression 1 month after CCH induction [44]. Nevertheless, a study using C57BL/6 mice reported that CCH increased Aβ levels and enhanced β/γ-secretase levels after 8 months of bilateral carotid artery stenosis [30]. The latest findings suggest that CCH in C57BL/6 mice could induce Aβ accumulation or promote Aβ pathogenesis only if CCH was applied for a long period. However, this hypothesis needs further investigation.

Increased HMGB1 at the acute phase of hypoperfusion induces persistent neuroinflammation [22]. However, in the present study, a remarkable increase in HMGB1 was observed during the chronic phase of CCH over 3 months. This might be a response of dying cells or surrounding cell death due to hypoxia and ischemia caused by CCH, which eventually leads to neuronal loss [3,45,46]. The increased HMGB1 during the chronic phase is not released from necrotic neurons because necrosis only occurs in the acute phase of CCH [6]. Instead, it is possibly released by apoptotic neurons because apoptosis is the main cause of neuronal death at the chronic phase of CCH [6,47]. As the role of HMGB1 in the chronic stage of CCH has never been explored, the increased HMGB1 level in the present study could be a novel finding to support the contribution of HMGB1 at the chronic phase of CCH. Moreover, it could serve as further evidence that HMGB1 is released by apoptotic neurons after CCH.

Once HMGB1 is released extracellularly, it binds to TLR2, TLR4, or RAGE receptors, then activating nuclear factor kappa-light chain-enhancer of activated B cells (NF-κB) pathway to initiate the production of proinflammatory cytokines, such as IL-1β, IL-6, and TNF-α [17,18]. As confirmed by our findings, TNF-α and IL-1β levels were higher in BCCAO mice compared with sham mice at 3 months after CCH. However, the level of TLR2/4, RAGE, or protein involved in the NF-KB signaling pathway did not increase (data not shown). Thus, HMGB1 triggered the downstream inflammation pathway without changing the levels of its receptors, highlighting the crucial role of HMGB1 in the VCI pathophysiology induced by CCH.

The administration of the HMGB1-CRISPR/Cas9 KO plasmid during the chronic phase of CCH suppressed HMGB1 and proinflammatory cytokines (TNF-α, IL-1β, and IL-6). Moreover, it successfully attenuated hippocampal atrophy and improved cognitive impairment at the chronic phase of CCH. Thus, HMGB1 and its proinflammatory cytokines may contribute to neuronal loss and cognitive impairment, as shown by hippocampal atrophy and NOR test, respectively.

The present study has some limitations. First, we investigated the effects of HMGB1 suppression only at a single dose and one period, that is, 2 months after CRISPR injection. This is a concept-proving study, and the effects of HMGB1 suppression at different doses and for different periods should be determined for clinical application. Second, we did not investigate the role of HMGB1 in glial cells. Furthermore, studies have shown that HMGB1 are released rapidly after cerebral ischemia by astrocytes or 1 week after ischemia by microglia [48,49]. Moreover, HMGB1 activates glial cells after it is released by dying neurons [18,50]. Therefore, future studies must investigate the role of HMGB1 in glial cells after CCH. Next, the underlying mechanisms by which HMGB1 exerts its proinflammatory action after CCH is unclear in this study. Studies have proven that HMGB1 binds to TLR2/TLR4 or RAGE, which was consistent in this study, initiating the downstream inflammatory pathway [51,52]. An investigation focused on molecular signaling underlying the increased HMGB1 expression at the chronic phase of CCH may be considered for future study.

## 4. Materials and Methods

### 4.1. Animals and Study Design

Male C57bl/6j mice (16 weeks old, weighing 25–35 g, Bio-Lasco Taiwan Co., Ltd., Taiwan) were used for all experiments. They were placed under controlled temperature (22  ± 1 °C) and humidity (55  ±  10%), with a 12-h light/dark cycle (lights on at 07:00 h). Food and water were given ad libitum to all mice throughout experiments. Animal care and experimental procedures in this study were performed in accordance with guidelines for the Care and Use of Laboratory Animals from Ethics Committee of Taipei Medical University. The animal use protocol had been reviewed and approved by the Institutional Animal Care and Use Committee or Panel (IACUC/IACUP) on 04 January 2017 with the approval No: LAC-2016-0434.

The groups of mice were divided into two: group 1 was for before HMGB1 CRISPR/Cas9 KO plasmid injection, and group 2 was for after the injection. Mice in group 1 were randomly categorized as sham-control and modified-BCCAO. Mice in group 2 were randomly categorized as sham-CRISPR control, sham-CRISPR HMGB1 KO, BCCAO-CRISPR control, and BCCAO-CRIPSR HMGB1 KO (Figure 6). This study followed the ARRIVE (Animal Research: Reporting of In Vivo Experiments) guidelines.

### 4.2. Model Establishment of CCH through Modified BCCAO Surgery

The modified-BCCAO was performed through ligation of both the common carotid artery (CCA) but with some modifications of the previous procedure [53,54]. Basically, the BCCAO procedure is usually performed in rats through ligation of the right and left common carotid arteries with silk suture. Both ligations are permanent and performed on the same day. This procedure is restricted to rats because they have a complete circle of Willis for brain vascularization. However, mice lack fully developed posterior communicating arteries of the circle of Willis, which connect the carotid and vertebral systems. Therefore, mice could die from severe ischemia if BCCAO is applied to them [6,55,56]. Hence, we modified the BCCAO procedure by performing the two-step surgery. The first step was ligating the right CCA permanently. The second step was performed 1 week after the first step through ligation of the left CCA transiently for 30 min. After 30 min, the ligation was removed to allow blood supply to revascularize the brain.

To permanently ligate the right CCA, mice were anesthetized with 2% isoflurane. CBF was measured while performing surgery. A sagittal midline incision (~1 cm in length) was made to expose the parietal skull. The skin was carefully dissected, and a fiber optic probe of laser doppler flowmetry was put directly into the skull 2 mm caudally and 5 mm laterally from the bregma to measure the CBF. A cervical midline neck incision (~1 cm in length) was made. Both the salivary glands were carefully separated and mobilized to visualize the underlying CCA. Both the CCA were carefully separated from the respective vagal nerves and accompanying veins without harming these structures. A tight, double 5-0 silk suture loop (proximal and distal) was made around the right CCA. The CBF remeasurement showed an 80–90% reduction compared with the baseline. Finally, we closed the wound with sutures.

After 1 week, the procedure was repeated for the left CCA. The steps were similar to those followed for the right CCA. However, the ligation was applied only transiently for 30 min. A small polyethylene tubing (diameter of 0.58 mm) was inserted between the left CCA and silk sutures. This tubing was used as a splinting of the left CCA to avoid damaging arterial walls when tightening the sutures. The left CCA was occluded for 30 min through tightening of silk sutures. Then, the CBF remeasurement showed reduction by 80–90% compared with the baseline. After 30 min, the ligation of the left CCA was released and the polyethylene tubing was removed. After the whole procedure, mice were placed in a heating pad for 30 min until they woke up. After awaking, they were placed back into their cages.

The procedure for sham surgery was the same as that used for BCCAO surgery. However, both common carotid arteries of mice were exposed without ligation.

### 4.3. Injection of HMGB1 CRISPR/Cas9-KO Plasmid

The HMG-1 Crispr/Cas9 KO plasmid (sc-400735) and HMG-1 homology-directed DNA repair (HDR) plasmid (sc-400735-HDR) were purchased from Santa Cruz Biotechnology, Inc, California, USA. The HMG-1 Crispr/Cas9 KO plasmid consisted of a pool of three plasmids, each encoding the Cas9 nuclease and the HMGB-1-specific 20-nt guide RNA (gRNA) designed for maximum KO efficiency. gRNA sequences were derived from the GeCKO (v2) library and directed the Cas9 protein to induce a site-specific double-strand break in genomic DNA. The HMG-1 HDR plasmid consisted of a pool of two to three plasmids, each containing an HDR template corresponding to cut sites generated by the HMG-1 Crispr/Cas9 KO plasmid. Each HDR template contained two 800-bp homology-arm designed to specifically bind to genomic DNA surrounding the corresponding Cas9-induced double-strand DNA break site.

We injected the HMG-1 Crispr/Cas9 KO plasmid and HMG-1 HDR plasmid mixed with jetSI^TM^ 10 mM transfection reagent (Polyplus Transfection, New York, USA) according to the manufacturer’s protocol. A total of 4 μL of the mixture was injected into each hippocampus (anteroposterior—2 mm, mediolateral—1.5 mm, and dorsoventral—2 mm related to the bregma) at a rate of 0.4 μL/min by using a 26-gauge Hamilton syringe under isoflurane anesthesia. For the CRISPR control group, we injected a vehicle with the same volume by using the same procedure. The injection was administered 1 month after BCCAO/sham surgery because this time is a subacute or transitional phase from the acute phase of CCH to the chronic phase [6]. Two months after the injection (or 3 months after BCCAO surgery), mice were examined through magnetic resonance imaging (MRI) to evaluate brain pathology and hippocampal atrophy as well as by using the novel object recognition (NOR) test. During this time, BCCAO mice were in a condition of chronic phase of CCH [6].

### 4.4. Novel Object Recognition

Mice tend to interact more with a novel object than with a familiar one. This tendency has been used by behavioral pharmacologists and neuroscientists to study learning and memory. A popular protocol for such research is the object recognition task [48]. The procedure consists of three phases: habituation, sample, and test. We modified this procedure [57]. During the habituation phase, each mouse was allowed to explore the field in the absence of objects for 10 min for two consecutive days to make them familiar with the field. On the third day, during the sample phase, two objects were placed symmetrically onto the arena. Mice were placed at the mid-point between the wall and sample objects, with their bodies parallel to side walls and their noses pointing away from objects, and they were allowed to freely explore for 5 min. Time spent exploring objects was recorded. During the test phase, one of the two objects used in the sample phase was randomly replaced by a novel one, and then, mice were re-introduced to the arena for 5-min exploration after a 4-h delay. Between each phase, any feces were cleared, and the arena and objects were cleaned with 70% ethyl alcohol. The video tracking system was used to collect behavioral performances automatically. The time spent exploring both novel and familiar objects was recorded (TN and TF, respectively). Object discrimination was evaluated through the recognition index (RI): RI = TN/(TN + TF).

### 4.5. Motor Function Test

Beam walking test: Further subtle motor coordination and balance were assessed using the modified balance beam (beam walking) test. This procedure was based on a modified protocol described by Luong et al. [58]. The beam apparatus consisted of 1-m beams with a round-rough surface of 12 mm (or 6 mm width) resting 50 cm above the table top on two poles. A black box was placed at the end of the beam as the finish point. Food was placed in the black box to attract the mouse to the finish point. A lamp (with a 60-watt light bulb) was used to indicate the start point and served as an aversive stimulus. The time to cross from the center to up to 80 cm was measured manually: the timer was started at 0 cm and stopped at 80 cm. A video camera was set on a tripod to record the experiment. This experiment was conducted over 3 consecutive days: 2 days of training and 1 day of testing. On training days, each mouse crossed the 12-mm (or 6-mm) beam three times. On the testing day, time taken to cross each beam was recorded. Two successful trials in which the mouse did not stall on the beam were averaged. Video recordings could be used for the finer analysis of slipping and other observable motor deficits.

Rotarod: Briefly, gross motor control was measured using the rotarod (IITC Life Science, CA, USA). This procedure was based on the protocol described by Tung et al. with some modifications [59]. For this test, each mouse was placed on a cylindrical dowel (69.5 mm in diameter) raised 27 cm above the floor of a landing platform. Mice were placed on dowels for 5 min to acclimatize them to the test apparatus. Once initiated, cylindrical dowels began rotating and accelerated from 5 rpm to a final speed of 44 rpm over 60 s. During this time, mice were required to walk in a forward direction on rotating dowels for as long as possible. When mice were no longer able to walk on rotating dowels, they fell onto the landing platform below. This indicated the end of the trial for an animal, and the time to fall was noted. Passive rotations where mice clung to and consequently rotated with the dowel were also defined as the end of the trial. Mice were then returned to their cages with access to food and water for 10 min. This procedure took 3 days; day 1 and 2 were for training (each day consisted of two to three trials), and day 3 was for testing (it consisted of two trials). The trials from the testing day were used for the analysis.

### 4.6. 7T-aMRI (7Tesla-Animal MRI)

After BCCAO, MRI was performed using a 7T horizontal MRI scanner (Bruker PharmaScan 70/16, Bruker Biospin, Billerica, MA, USA), 7T/40 cm magnet (Biospect Bruker console), and a surface coil to monitor the brain condition (e.g., atrophy and white matter lesion). Mice were initially anesthetized with 3.0% isoflurane (Escain, Mylan Japan, Tokyo, Japan) and then with 1.5–2.0% isoflurane and 1:5 oxygen/room-air mixture during MRI experiments. Rectal temperature was continuously monitored using an optical thermometer (FOT-M, FISO, Quebec, QC, Canada) and maintained at 37.0 °C ± 0.5 °C by using a heating pad (Temperature control unit, Rapid Biomedical, Torrington, CT, USA), and warm air was provided by a homemade automatic heating system based on an electric temperature controller (E5CN, Omron, Kyoto, Japan) throughout MRI experiments. During MRI scanning, mice were laid in a prone position on an MRI-compatible cradle and were held in place with handmade ear bars. The first imaging slices were carefully set at the rhinal fissure, with reference to a mouse brain atlas. The modality of MRI performed was T1-weighted fast low angle shot image and T2-weighted spin echo.

Trans-axial T1-weighted fast low angle shot image was acquired sequentially as follows: TR/TE = 341/4.5 ms, slice number/thickness = 16/0.75 mm, matrix = 256 × 256, FOV = 16 × 16 mm^2^, average = 8, flip angle = 30, and scan time = 8 min 44s.

Trans-axial T2-weighted images were acquired using rapid acquisition with a relaxation enhancement (RARE) sequence as follows: TR/TE = 2500/33 ms, slice number/thickness = 16/0.75 mm, matrix = 256 × 256, FOV = 16 × 16 mm^2^, average = 8, RARE factor = 8, flip angel = 90, and scan time = 10 min 40s. Volumes of the hippocampus and lateral ventricles were measured using MRIcron software (NITRC 2016, University of South Carolina, Columbia, SC, USA).

### 4.7. Amyloid Positron Imaging Tomography Scanning

For amyloid positron imaging tomography (PET) scanning, mice were anesthetized with isoflurane (1.5%, delivered through a mask at 3.5 L/min in oxygen) and received bolus injection of 18.5 MBq/0.2 cc of ^18^F-AV45 (Eli Lilly, Indianapolis, IN, USA) through the tail vein with a catheter. Using a scanner (Inveon, Siemens Medical Solutions, Munich, Germany), a 10-min transmission scan was obtained with a rotating ^57^Co point source, followed by a single-frame emission recording for the interval 40–60 min post-injection. The PET image reconstruction procedure consisted of a three-dimensional ordered subset expectation maximization with four iterations and 12 subsets followed by a maximum a posteriori algorithm. Scatter and attenuation correction were performed, and a decay correction for ^18^F was applied. With a zoom factor of 1.0 and a 128 × 128 × 159 matrix, a final voxel dimension of 0.78 × 0.78 × 0.80 mm was obtained. The image was analyzed using PMOD (PMOD version 3.7, Technologies Ltd., Switzerland). A manual rigid-body re-alignment of individual ^18^F-AV45 images on an ^18^F-AV45 template was performed using the PMOD fusion tool. The normalized PET images were co-registered to an MRI brain template for region of interest delineation. Using the predefined mouse brain MRI template of PMOD and manual ROI of the frontal lobe and hippocampus fused with co-registered AV-45 PET/CT image, the AV-45 signal in the region of interests was determined, respectively.

### 4.8. Western Blot

At 3 months after BCCAO or sham surgery, mice were anesthetized with 2% isoflurane and then euthanized. The brains were collected from the skull, and each hemisphere was separated into three regions: cortex, hippocampus, and striatum. As the hippocampal tissue was small, we combined the right and left hippocampus for the Western blot experiment. Protein samples of each region were extracted in the following manner. Individual tissue samples were homogenized in lysis buffer (50 mmol/L Tris, pH 7.4; 1 mmol/L ethylenediaminetetraacetic acid; 1 mmol/L phenylmethylsulfonyl fluoride; 4 μg/ml aprotinin and leupeptin; and 1% sodium dodecyl sulfate), protease inhibitor cocktail solution, and phosphatase inhibitor cocktail solution (GenDEPOT, Barker, Katy, TX, USA). The homogenates were then centrifuged at 13.400 × g for 30 min at 4 °C, and supernatants were harvested, snap-frozen, and stored at −80 °C. The protein concentration of supernatants was determined using the Bradford assay. Equal amounts of protein (20 μg) were then separated through sodium dodecyl sulfate–polyacrylamide gel electrophoresis and transferred to a polyvinylidene difluoride membrane, which was subsequently incubated in a primary Ab against HMGB1 (mouse monoclonal Ab, 1:1000, GeneTex, Irvine, CA, USA), interleukin (IL)-1α (rabbit polyclonal Ab, 1:1000, GeneTex, Irvine, CA, USA), tumor necrosis factor-alpha (TNF-α; rabbit polyclonal Ab, 1:1000, GeneTex, USA), and IL-6 (rabbit polyclonal Ab, 1:1000, GeneTex, Irvine, CA, USA). Following incubation in a primary Ab, membranes were incubated in horseradish peroxidase-conjugated secondary Ab (Cell Signaling, Danvers, MA, USA) and then detected using an ECL system (Thermo Scientific, Waltham, MA, USA) with a UVP Biospectrum AC system (Fisher Scientific, Pittsburgh, PA, USA). Densitometry is performed for specific markers normalized to β-actin (1:1000, clone C4, Millipore Corporation, Temecula, CA, USA) by using Image J (v1.37) software.

### 4.9. Immunostaining

At 3 months after BCCAO or sham surgery, mice were anesthetized and intracardially perfused with phosphate-buffered saline (PBS), followed by 4% paraformaldehyde. The brains were removed, post-fixed for 4 h in 4% paraformaldehyde at 4 °C, and stored in 30% sucrose in 0.1 M PBS (pH 7.4). Serial coronal-cryopreserved sections of 30-μm thickness that spanned from the anterior of the corpus callosum (bregma, 0.26 mm) to the anterior of the hippocampus (bregma, 0.94 mm) (adjusted according to the mouse brain atlas) [60] were used as a cryostat. The slides were then permeabilized with 0.5% Triton X-100 in PBS for 2 h. After blocking with BlockPRO (VISUAL PROTEIN) at room temperature for 1 h, the brain slides were incubated with the following primary antibodies: mouse monoclonal anti-HMGB1 Ab (1:100, GeneTex, Irvine, CA, USA), rabbit polyclonal anti-TNF-á (1:100, GeneTex, Irvine, CA, USA), rabbit polyclonal anti-IL-6 (1:100, GeneTex, Irvine, CA, USA), and rabbit polyclonal anti-IL-1β (1:100, GeneTex, Irvine, CA, USA) at 4 °C overnight. The brain slides were then incubated with secondary antibodies conjugated to Alexa 488 (1:1000; Thermo Fisher Scientific, Waltham, MA, USA) or Alexa 594 (1:500; Thermo Fisher Scientific, Waltham, MA, USA) for 2-h at RT. The brain sections were then washed and counterstained with 2 μg/mL 4′,6-diamidino-2-phenylindol (Life Technologies, Carlsbad, CA, USA) for 20 min at RT. Fluorescence images were obtained using a Tissue FAXS system (TISSUE GNOSTICS, North America Company, Los Angeles, CA, USA) with a ×25 objective lens. For the triple stain, tissue slides were incubated with primary antibodies: mouse monoclonal anti-HMGB1 Ab (1:100, GeneTex, Irvine, CA, USA) and rabbit polyclonal anti-NeuN (1:300, Cell Signaling, Danvers, MA, USA). Images were obtained using the confocal microscope with ×63 oil objective lens. All images were obtained at 1024 × 1024-pixel resolution.

### 4.10. Statistical Analysis

One-way analysis of variance (ANOVA) followed by Sidak’s multiple comparison tests were used to analyze Western blot results, IF, and CBF measurement. NOR results were analyzed using two-way ANOVA. Student’s unpaired t-test was used to analyze motor function and MRI results. All analyses were performed using GraphPad Prism 7 software (GraphPad Software, La Jolla, San Diego, CA, USA). Statistical significance was considered at *p* value < 0.05.

## 5. Conclusions

This is the first study to demonstrate that HMGB1 significantly contributes to cognitive impairment at the chronic phase of CCH, partly through inflammation modulation. The HMG-1 CRISPR/Cas9 KO plasmid reversed cognitive impairment. Thus, HMGB1 suppression might be a promising therapeutic method for VCI due to CCH.

## Figures and Tables

**Figure 1 ijms-21-02176-f001:**
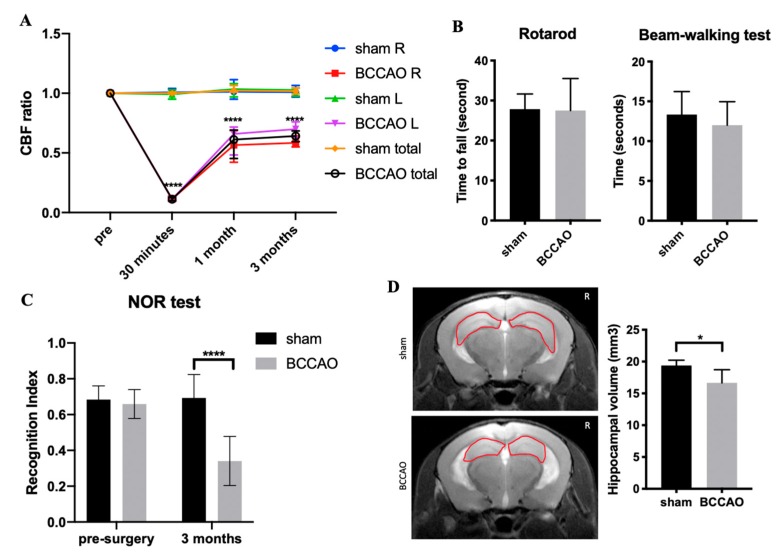
Effect of chronic cerebral hypoperfusion (CCH) on mice at 3 months after the bilateral common carotid artery occlusion (BCCAO) and sham surgery. (**A**) Cerebral blood flow (CBF) reduced to 50–70% from the baseline in BCCAO mice; (**B**) CCH did not alter motor coordination, which was observed as no difference in the rotarod and beam-walking test; (**C**) Novel object recognition (NOR) test showed that BCCAO mice had memory decline compared with sham mice after CCH induction; (**D**) CCH induced hippocampal atrophy in BCCAO mice (the area of hippocampus is shown by a red circle). *n* = 6 for each group. Data are presented as the mean + standard deviation; * *p* < 0.05 by unpaired t-test (hippocampal volume); **** *p* < 0.0001 by two-way analysis of variance followed by Sidak’s multiple comparison test (CBF ratio and NOR). Sham R: right brain area of sham mice, sham L: left brain area of sham mice, BCCAO R: right brain area of BCCAO mice, BCCAO L: left brain area of BCCAO mice.

**Figure 2 ijms-21-02176-f002:**
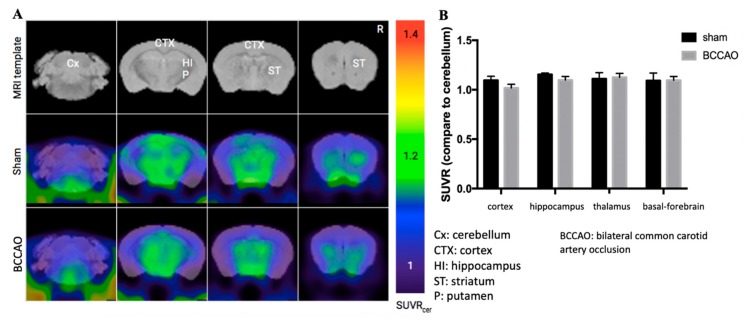
Chronic cerebral hypoperfusion induced by modified bilateral common carotid artery occlusion (BCCAO) did not cause amyloid-beta accumulation after 3 months. (**A**) Amyloid positron imaging tomography scan image. (**B**) Quantification data of Aβ accumulation were same for sham and BCCAO mice in some brain regions. *n* = 6 for each group. Data are presented as the mean ± standard deviation based on one-way analysis of variance.

**Figure 3 ijms-21-02176-f003:**
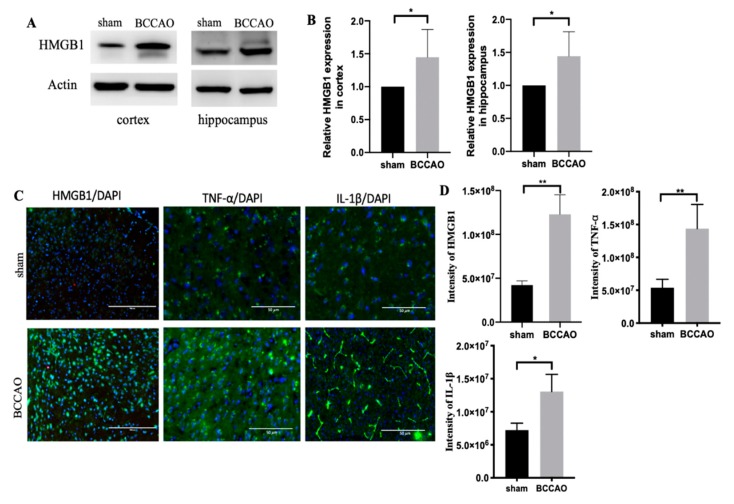
Increased high-mobility group box protein 1 (HMGB1) and its proinflammatory cytokines (tumor necrosis factor-alpha [TNF-α] and interleukin [IL]-1β) in the cortex and hippocampus at 3 months after chronic cerebral hypoperfusion (CCH). (**A**) Western blot results of HMGB1 in the cortex and hippocampus. (**B**) Quantification results of HMGB1 relative protein in the cortex and hippocampus. (**C**) Immunostaining of HMGB1, TNF-α, and IL-1β in the cortex. (**D**) Quantification data of immunostaining image. *n* = 6 for each group. Data are presented as the mean + standard deviation; **p* < 0.05; ***p* < 0.01 with unpaired t-test; scale bar = 50 μm.

**Figure 4 ijms-21-02176-f004:**
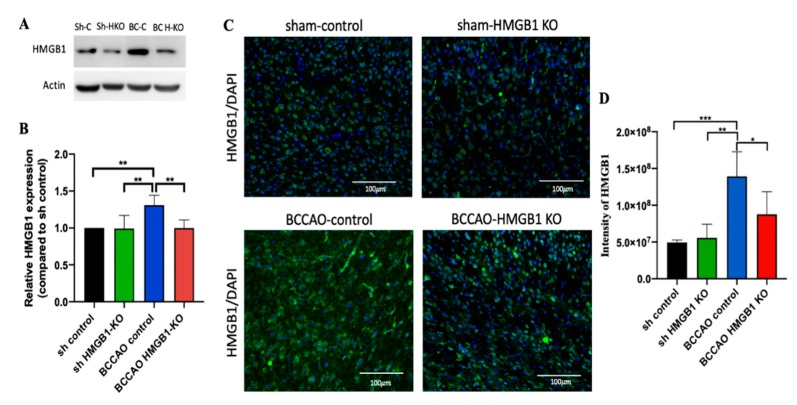
After injection of the HMG-1 CRISPR/Cas9 knockout (KO) plasmid, high-mobility group box protein 1 (HMGB1) expression in the cortex and hippocampus was suppressed, as confirmed through Western blot and immunostaining. (**A**) Western blot showed a decrease in HMGB1 expression in bilateral common carotid artery occlusion (BCCAO) HMGB1-KO mice compared with BCCAO controls. (**B**) Quantification data of Western blot results. (**C**) Immunostaining further confirmed the Western blot finding that HMGB1 expression decreased in the cortex of BCCAO HMGB1-KO mice but not in BCCAO controls. (**D**) Quantification data of immunostaining image. *n* = 5 for each group. Data are presented as the mean + standard deviation; **p* < 0.05; ***p* < 0.01; ****p* < 0.001 with one-way analysis of variance followed by Sidak’s multiple comparison test; scale bar = 100 μm. Sh-C: sham control, Sh-HKO: sham HMGB1 KO, BC-C: BCCAO control, BC-HKO: BCCAO HMGB1 KO.

**Figure 5 ijms-21-02176-f005:**
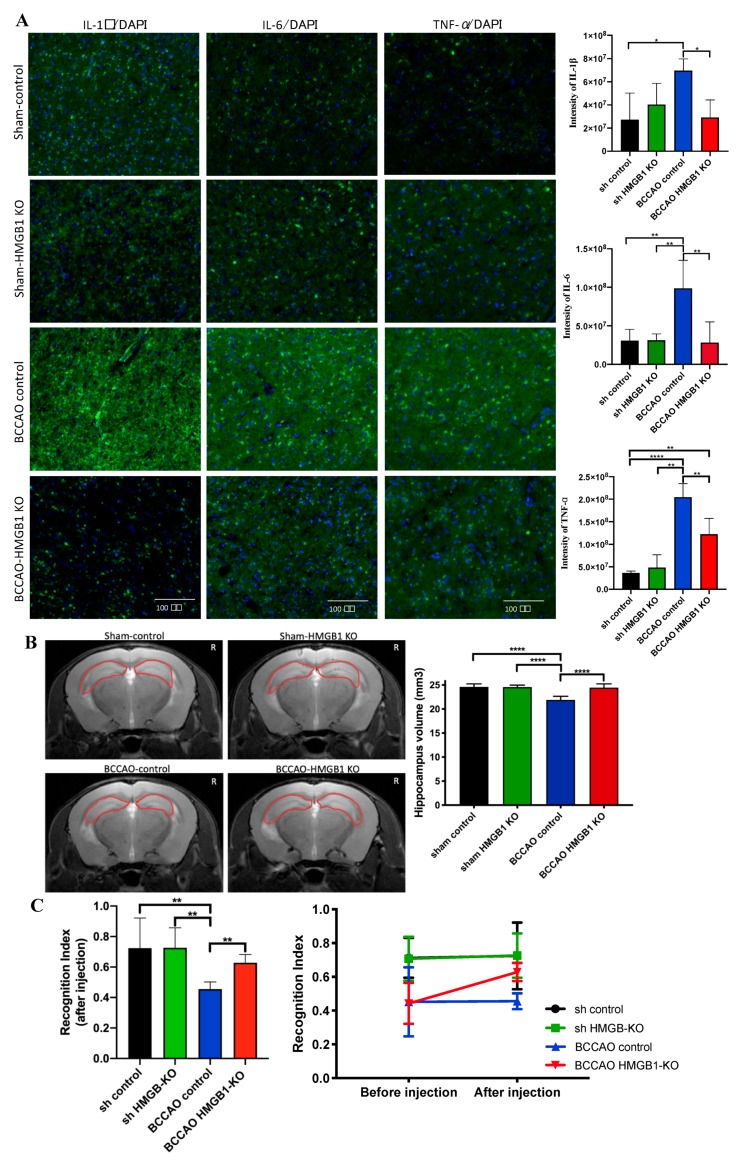
The effect of high-mobility group box protein 1 (HMGB1) suppression with CRISPR/Cas9 knockout (KO) plasmid on proinflammatory cytokines, hippocampus volume, and memory function. (**A**) Decreased level of tumor necrosis factor-alpha, interleukin (IL)-1β, and IL-6 of bilateral common carotid artery occlusion (BCCAO) HMGB1-KO mice showed with immunostaining. (**B**) Attenuation of hippocampal atrophy in BCCAO HMGB1-KO mice but not in BCCAO controls (the area of the hippocampus is indicated with a red circle). (**C**) Improvement of memory decline in BCCAO HMGB1-KO mice but not in BCCAO controls. *n* = 5 for each group. Data are presented as the mean + standard deviation; **p* < 0.05; ***p* < 0.01; ****p* < 0.001 with one-way analysis of variance (ANOVA) followed by Sidak’s multiple comparison test; novel object recognition test before and after CRISPR/Cas9 injection with two-way ANOVA (F(3,24) = 7.598, *p* < 0.01); scale bar = 100 μm.

**Figure 6 ijms-21-02176-f006:**
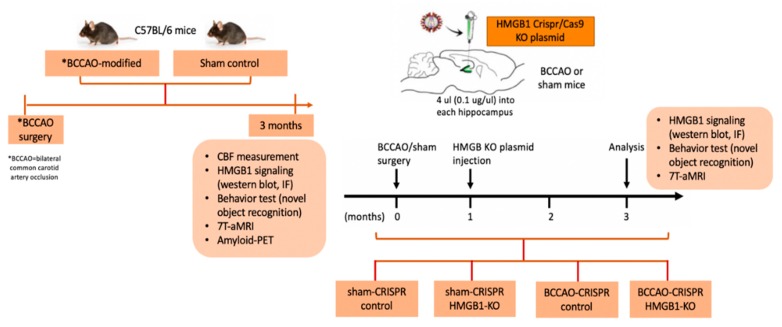
Schematic description of the experimental design.

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
