# Peer review of "Role of HMGB1 in an Animal Model of Vascular Cognitive Impairment Induced by Chronic Cerebral Hypoperfusion"

_ijms, 2020, doi:10.3390/ijms21062176_

Round 1

Reviewer 1 Report

I think this study is theoretical and the methods used are appropriate. And I am satisfied with the authors' reply to my comments.

Reviewer 2 Report

Thank you for making the necessary changes and for describing your modified method in detail as requested.

The procedure is now well described and your reasons to make modifications are now clear to the reades.

Some minor language editing could still improve your manuscript. The changes are, however, already sufficient and the work is sound.

This manuscript is a resubmission of an earlier submission. The following is a list of the peer review reports and author responses from that submission.

Round 1

Reviewer 1 Report

This study shows that HMGB1 appeared in the chronic CCH is contributed to the production of the pro-inflammatory cytokines and the cognitive impairment using chronic CCH model mice and HMGB1-CRISPR/Cas9 KO plasmid. I think this story is theoretical and they use the appropriate methods. I have some minor concerns for the readers.

L47: “decreased” may be decrease”. Fig.2: Abbreviations such as BCCAO R and BCCAO L used in the figure should be described in the legend. In Fig.2-D, red painted might prevent the reader to realize the brain area, so it should be drawn by the dotted line or something. L322: Striatum isn’t in Fig4, so it should be omitted in this sentence. L331~333: Fig.6A shows the significant increase of IL-6 in BCCAO mice? Fig6B: The red parts in the figure are the same as those in Fig2D.

Reviewer 2 Report

This paper discusses vascular cognitive impairment and chronic cerebral hypo perfusion, which is important to further understand due to the number of Alzheimer's patients in the world. One of the major causes of decline in these patients is chronic cerebral hypo perfusion. HMGB1 is a cytokine produced in response to infection, injury and inflammation and leads to neuronal damage and worsening symptoms. After the acute phase of chronic cerebral hypoperfusion, the role of HMGB1 is not well defined, which was the goal of this investigation.

Line 24 - HMGB1 needs defined

Line 28-29 needs grammatical correction.

Line 95: BCCAO needs defined

Lines 102-103 needs grammatical correction

The methods are very confusing.It is described initially that only one of the common carotid arteries was ligated transiently for 30 minutes. However, it is described that one artery was ligated permanently and then a week later the other may have been partially occluded? This is not very clearly described and should be revised.

Lines 139-141: why was the injection done 1 month after surgery and then the MRI 2 months later? Why was this timing done?

What were the number of animals used for each experiment? For example, how many mice had MRI, PET scanning, Western blot and immunostaining completed? The n's are not listed in the results either, which is very concerning that only 1-2 mice were used for the experiments. What sexes were used?

Figure 2: The MRI images used do not appear to be the same location; one is slightly more anterior than the other. Are there better examples to use?

Reviewer 3 Report

General comment

This is an excellent work on the role of HMGB1 in cognitive impairment induced by cerebral hypoperfusion. Additional english language editing is strongly advised.

Specific comments

Page 3, line 100 ff: Please rephrase this section to clarify the details of the modified technique used.

Page 6, line 272: for a waiting time
